

# Odd versus even: a scientific study of the 'rules' of plating

Andy T. Woods, Charles Michel and Charles Spence

Crossmodal Research Laboratory, Department of Experimental Psychology, University of Oxford, Oxford, United Kingdom

## ABSTRACT

We report on the results of a series of large-scale computer-based preference tests (conducted at The Science Museum in London and online) that evaluated the widely-held belief that food should be plated in odd rather than even numbers of elements in order to maximize the visual appeal of a dish. Participants were presented with pairs of plates of food showing odd versus even number of seared scallops (3 vs. 4; 1–6 in Experiment 7), arranged in a line, as a polygon or randomly, on either a round or square white plate. No consistent evidence for a preference for odd or even numbers of food items was found, thus questioning the oft-made assertion that odd number of items on a plate looks better than an even number. The implications of these results are discussed.

## INTRODUCTION

*"Aller guter Dinge sind drei, nicht vier"*—*all good things come in threes not four* (popular German saying).

Disciplines that involve arranging items to maximize their aesthetic appeal share the common aim of seducing the eyes of the observer. From gardeners to chefs—plants and rocks at one end, to food elements on the plate on the other—each discipline has its own insights concerning ways to enhance composition. These ideas are often transmitted orally, or sometimes, via books or guides. We believe that while the approach has historically been driven by experience, studying which of those guidelines are effective (and which of those that turn out to be effective cross-culturally) could benefit both these artisanal (or, may we say, artistic) disciplines, but also pose interesting scientific questions as to the nature of those biases, should they be confirmed empirically.

One such guideline is the belief that it is better to present odd rather than even numbers of items (e.g., *Van Tonder & Lyons, 2005*).[1] But is this anything more than 'an old wives' tale? Chefs often acknowledge the importance of presenting odd numbers of

Corresponding author
Andy T. Woods,
andytwoods@gmail.com

---

[1] The idea of a preference for odd rather than even numbers of elements is not new. In the traditional art of Japanese rock gardens, for instance, ancient texts mention the importance of preferring odd vs. even numbers (*Shimoyama, 1976*; *Van Tonder & Lyons, 2005*), not only in terms of odd-numbered groups of rocks, but also that the different clusters of rocks should also be an odd number.

[2]An eagle-eyed reviewer raised the possibility that an odd number of food-items could be preferred over an even number due to reasons of etiquette. Their example was that, "given the assumption that food on a dish might be consumed only partially (and this would very much apply to a context of cuisine rather than kitchen), an odd-item presentation would favor a division with remainders."

elements on the plate, as recommended in chefs' guides on the art of plating (e.g., *Styler & Lazarus, 2006*), in an attempt to enhance the visual appeal of a meal.[2] However, to the best of our knowledge, this claim has not been empirically tested previously. Therefore, the question that we wish to address here is: "Do odd vs. even numbers really matter when it comes to the visual appreciation of compositions?" And, to what extent can this difference influence the visual appreciation of the food, or maybe even the actual enjoyment of the food? One way of testing whether an odd number of items on a plate is preferred over an even number is to show participants two such plates of food, and ask them which they prefer. Unfortunately, any two such plates of food would undoubtedly differ in more ways than just the number of items that they contain, which makes it hard to tease out the underlying drivers of liking. We discuss these issues shortly.

In terms of food, there is very little research on the topic. Furthermore, none of this research has focused on the question of odd versus even numbers. Bajaj was one of the only researchers to tangentially address this issue. In his doctoral thesis, 215 participants were given the option of eating a piece of chicken cut into either 4 pieces, or left as a single piece (*Bajaj, 2013*, Chapter 3 Experiment 1). Although significantly more participants chose the 4-item dish over the 1-item dish than could be expected by chance (148 vs. 67, $p < .001$), no difference in pleasantness was reported between these individuals and those deciding on the 1-item dish. In a second study, 301 participants were randomly assigned to meal type (a bagel served in 4 pieces vs. whole) but pleasantness did not vary across the groups of participants. The issue with these studies, in relation to 'odd versus even' number of items on the plate, is that the number of food items were quite different (1 vs. 4). We would expect, and will discuss next, a range of issues that might have swayed one's opinion on dish preference, which most likely are only exacerbated by large differences in the number of items/sub-portions.

For example, *Geier, Rozin & Doros (2006)* put forward, and subsequently demonstrated, the concept of 'unit bias,' where, when given the option to eat to satiation items of a small or large size, much smaller quantities of the small items were consumed than of the large. The consequence could be that, when asked to choose between plates of food, the most appetising portion will be that which matches one's current level of hunger (or dieting ambitions; see *Forde, Almiron-Roig & Brunstrom, 2015* for a recent review on expected sensation in food selection). The logical consequence for preference between odd versus even numbers of items on a plate is that, if one portion appears larger than the other, this may well have a knock-on effect on choice selection.

However, even if portions are equated in terms of their calorific content, a variety of phenomena can act to influence just how large a given portion of food may seem. For example, the size of the plate in relation to the food it contains has also been shown to influence perceived portion size thanks to the Delboeuf illusion (see *McClain et al., 2014*; *Spence et al., 2014*). This illusion occurs when circles placed within a surrounding circle are thought of as larger than they actually are when there is a small size difference between the circles, but smaller than they actually are if the size difference is larger.

The visual balance of the composition can influence how we perceive and how much we like food (for an overview, see *Spence & Piqueras-Fiszman, 2014*; C Michel et al., 2015,

unpublished data) and can presumably influence whether odd or even number of items on the plate are preferred. In terms of balance, *Zellner et al. (2011*, p. 642) state that: "*The presentation of a plate of food can be thought of as 'balanced' if that plate of food looks like it would balance when placed on a narrow central pedestal. That is, the food is distributed in a manner around the central point such that the perceived heaviness in one area looks balanced by equal heaviness on the opposite side of the plate.*" *Zellner et al. (2010)* found that balance, in conjunction with food colour (or lack of it), influenced the attractiveness of the visual presentation.

The artistic principles of visual harmony, including balance, contrast, emphasis, pattern, proportion, rhythm, unity, and variety (*Arnheim, 1988*; *Bouleau, 1980*; *Wilson & Chatterjee, 2005*), could also influence food preference (*Spence & Piqueras-Fiszman, 2014*). Some aspect of harmony could help to determine whether one prefers an odd versus an even number of items on the plate. Indeed, muddying the issue somewhat, the plate on which the food is presented could itself play in important role (as the 'frame' of the food).

## Overview

We report on a series of experiments that are currently running at the Science Museum in London (see Experiment 1 citizen science experiment). Participants were presented with photos of pairs of plates of food and asked to choose which one they preferred. The pairs always consisted of individual dishes of food, one containing an even number of seared scallops and the other an odd number of the same food. We also assessed any interaction between the odd/even, arrangement of the elements (line vs. polygon), and the shape of the plate on which the food elements happened to be presented.

The results of our first study revealed an intriguing interaction between odd/even and the shape of the plate on which the elements were arranged. There was, however, no consistent evidence for our hypothesis that 3 items would be preferred to 4 items. We explored these effects over a series of follow-up studies conducted online through Amazon's Mechanical Turk (MTurk). We controlled for the effects of crowding on the plate (Experiment 2), we equated portion size across the dishes (Experiments 3 and two further experiments reported as Supplemental Information), and we also tested for effects of portion size distortion (Experiment 4). The results of a Combined Analysis revealed that it was portion size that was the driving factor for both the participants at the Science Museum and those recruited via MTurk. These two groups of participants differed, though, in terms of which dish (odd versus even) they preferred when portion size was equated over plates. Whilst the participants in the Science Museum study appeared to prefer 3 items at this 'equal portion-size' point, the MTurk participants preferred 4 items. In Experiment 5, we ruled out the possibility that this difference was attributable to a small difference in the overall size of the two portions. We tested a third group of participants recruited through Prolific Academic in Experiment 6 to determine whether this group would have yet another equal portion size-point, but this was not the case. That is, the values obtained from this group did not really differ from that of MTurk participants. We argue, though, that the ratio-effect most likely arises due to some difference in the characteristics of the populations tested. In Experiment 7 we tested plates containing a range of numbers of elements, all of which

though were of the same portion size, and found that plates with more elements were generally preferred over those with fewer elements. Whether the dishes contained an odd or even number of elements played no role in this finding.

## EXPERIMENT 1: CITIZEN SCIENCE STUDY

Here we tested the hypothesis that participants would prefer a dish of food containing 3 items of food over one containing 4 items.

### Materials and methods

**Participants**

1,816 individuals (1,305 female and 509 male; 2 did not report whether they were male or female) took part in a citizen science experiment, conducted at the Science Museum in London during February to April 2015. The experiment could either be performed online (598 individuals)[3] or in an interactive digital platform at the 'Antenna Gallery,' as part of an exhibition on the science of eating called 'Cravings.' The online participants were invited to access this experiment via the information page of the 'Cravings' exhibition, and from the Science Museum's home webpage. At the museum's gallery, the digital platform was one of the attractions of the exhibition.

The median age of the participants was in the 16–34 years range (note that the participants specified if there age was <16, 16–34, 35–54, 55–74 or 75+; the respective counts in each group were 447, 880, 383, 92 and 12; 2 people did not report their age). All of the participants were informed about the nature of the study, and provided informed consent prior to taking part in the study and all of the studies reported thereafter. These studies have been approved by Oxford University's Medical Sciences Inter-Divisional Research Ethics Committee (approval # MSD-IDREC-C1-2015-004).

**Stimuli**

Scallops were chosen for the study, given that they are similar in shape (round) and size. Fresh scallops were seared in a hot pan with butter, in order to attain a light brown colouring. The same set of scallops was then placed and photographed on a white surface. Note that the scallops were photographed from a zenithal perspective with zenithal lighting, in order to avoid any shadow on the food. The scallop images were then cut and placed digitally on the different plates (square or round, photographed in the same way as the scallops). The stimuli used in this experiment are shown in Fig. 1.

**Design**

The dependent variable was the preferred dish chosen by the participants.

**Procedure**

The participants who took part in this experiment undertook five or more different tasks. The order in which the tasks were presented and the different conditions was randomised, as were the left or rightward position of the dishes. In the experiments reported here, 164 participants undertook two trials whilst the remainder completed only a single trial. The participants could either submit their answer by clicking on a circular button placed

[3]This experiment runs from the 20th of February 2015, until January 2016, see http://bit.ly/1MwGh35 to access the online experiment.

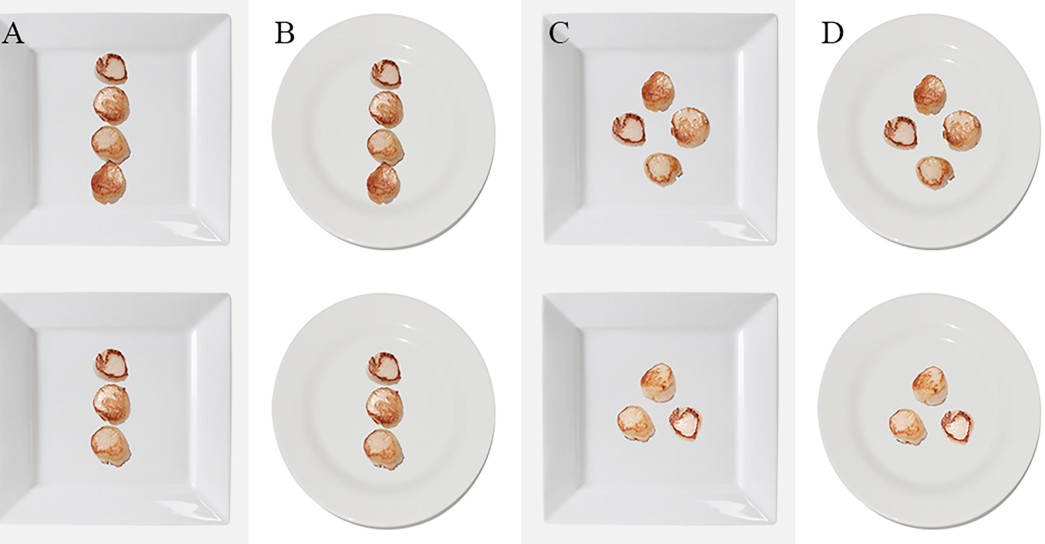

**Figure 1** **The 8 plates of scallops that were presented to the participants in Experiment 1.** The plates were presented in pairs (specifically, the upper and lower image in each column was compared). The plates vary systematically in terms of the number of seared scallops (3 vs. 4), the arrangement of the scallops (line vs. polygon), and the shape of the plate (round vs. angular).

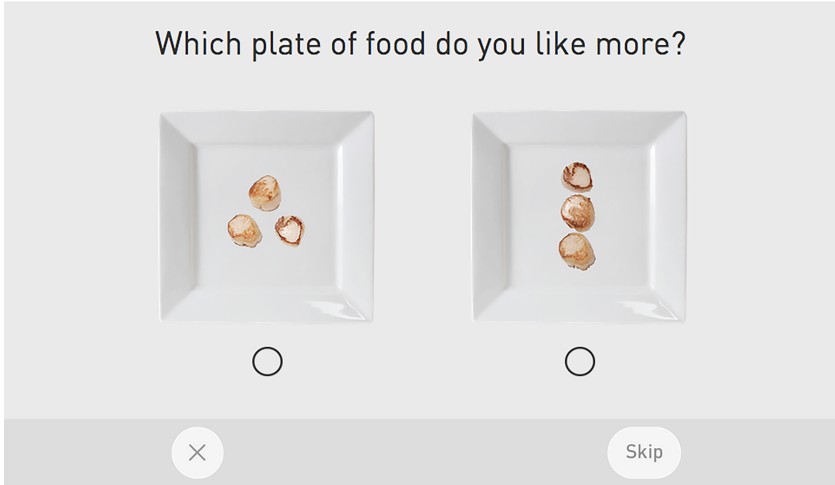

**Figure 2** **The arrangement of the scallops shown to participants on a single trial.**

right below the food image, leave the experiment by clicking on an 'X' button, or go on to the next question by clicking on the 'Skip' button (see Fig. 2).

## Results

The results, split by condition, are shown in Fig. 3. A log-linear analysis was performed, using Plate Shape (circular, square) × food Arrangement (vertical, polygonal) × food Items (3, 4) as the variables (the final model's likelihood ratio was $\chi^2(2) = 3.27, p = .20$). The Arrangement × Items $\chi^2(1) = 54.84, p < .001$, and Plate × Items interactions were retained by the model, $\chi^2(1) = 6.63, p = .01$. Both interactions were explored by means

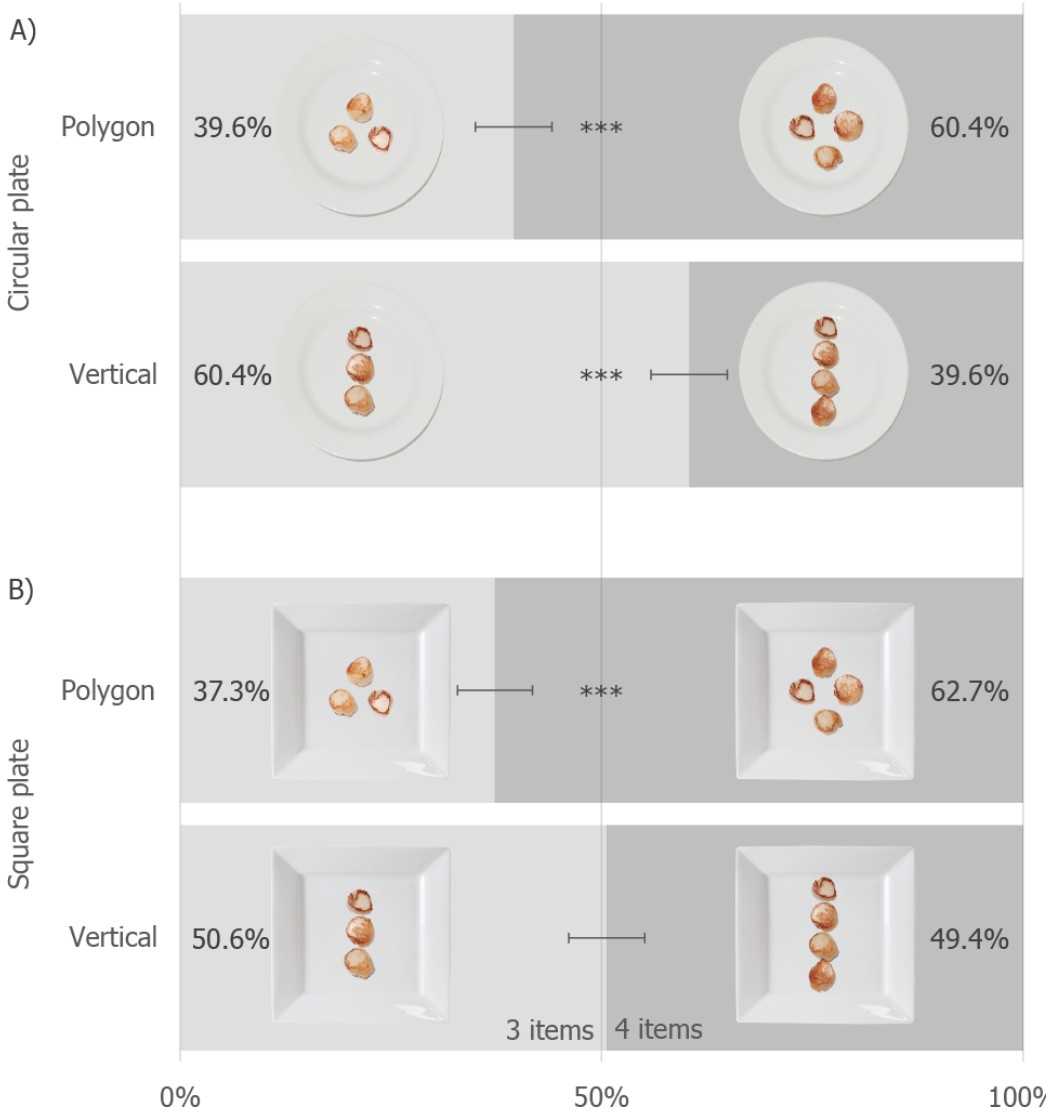

**Figure 3** The percentage of people preferring one dish over the over for each of the Plate and Arrangement conditions (error bars are 95% CI, \*\*\* = $p < .001$). The light grey shading representing preference for the 3-item dishes, and the dark grey the preference for the 4-item dishes.

of separate follow-up Exact Binomial tests designed to assess whether 3 vs. 4 items differed over the levels of the interacting factor.

In terms of the first interaction, in line with our hypothesis, 3 items that were arranged vertically were 1.24 times more likely to be chosen that 4 vertically arranged items ($p < .001$; with 428 picking the 4 item dish and 531 picking the 3 item dish; 95% CI [52.16%–58.55%]). Contrary to the hypothesis however, 4 items arranged as a polygon were 1.60 times more likely to be picked than 3 items arranged so ($p < .001$; with 578 picking the 4 item dish and 361 picking the 3 item dish; 95% CI [58.36%–64.68%]). In terms of the Plate × Items interaction, 4 items on a Square plate were 1.24 times as likely to be chosen over 3 items on a square plate ($p < .001$; with 540 picking the 4 item dish and

426 picking the 3 item dish; 95% CI [52.70%–59.06%]). There was no preference when it came to the Circular plates (466 picking the 4-item dish and 466 picking the 3-item dish; 95% CI [46.74%–53.26%]).

## Discussion

The findings do not generally support the hypothesis that dishes with an odd number of items would be preferred over dishes with an even number of items. Although our analysis did demonstrate that 3-vertically orientated scallops were preferred over 4-vertically orientated scallops, visual inspection of Fig. 3 shows that this effect only differed from that expected by chance when the scallops were plated on a circular plate. Thus, support for the hypothesis is actually more tenuous than that offered by the analysis. Indeed, overall, more evidence was found for 4 items being preferred over 3 items.

The lack of support for the hypotheses was unexpected, and after querying social media, several explanations were offered. One of the explanations proffered was that the portion sizes on 4-item plates were always seen as larger than those on 3-item plates. We tested for this in Experiments 3–6 by varying portion size by means of scaling the images of the scallops.

Two other issues were also suggested via social media. The first was that the four vertical items looked like substantially more food compared to those same number of items arranged as a polygon, and thus the dish was not preferred over the 3-item vertical dish as there was too much food on the plate. We go on to test this in Experiment 5 by asking the participants how hungry they were, and testing whether this influenced the results. There was, however, no evidence for such an effect.

The second more subtle issue was that the 4-item vertical dish looked a little less elegant to us than the vertical 3-item dish, perhaps as the plate was seen as being too full (some on social media even argued that the shape of the plate was distorted, becoming more oval). To test for this, in the next study, participants were exposed to dishes that were substantially larger than those used here, thus preventing the dishes from seeming too full.

## EXPERIMENT 2: TESTING FOR A CROWDED PLATE EFFECT

In this experiment we tested the hypothesis that participants found the plate crowded for the vertically arranged dishes, which influenced how participants decided between a 4-item vs. a 3-item dish. To do this, we conducted a similar study with the same factors as the previous (number of items, food alignment, and plate) and included an additional factor of plate size, albeit using a repeated measures design. Specifically, besides the 'regular' sized plate used in the previous study, we also collected data from those trials where a much larger plate was used instead.

## Materials and Methods

### Participants

One hundred participants[4] (35 female) were recruited from Amazon's Mechanical Turk to take part in the experiment in return for a payment of .40 US dollars. The participants

[4]A power analysis of the ratio of 3-item to 4-item preference for circular plated vertical/polygonal food from Experiment 1 (Generic Binomial Test, using G*Power 3.1.9.2) revealed that 90% power could be achieved in this study with an n of 62 or 64 (the former, vertically orientated food, the latter, polygonally-orientated food). We increased this to a sample size of 100.

ranged in age from 19 to 59 years ($M = 32.0$ years, SD $= 8.4$). Only participants recorded on MTturk as originating from the United States or Canada could take part in this and all subsequent MTurk studies. The experiment was conducted on 6/06/2015, from 12:00 GMT onwards, and over a two-hour period. The participants took an average of 73 s (SD $= 52$) to complete the study. All of the participants (here, and in subsequent studies) provided their informed consent prior to taking part in the study.

### Stimuli

The 5 unique scallop stimuli used in Experiment 1 were divided into separate transparent PNG files, as were the 2 plate stimuli. The 5 scallop images were individually resized so that they all contained approximately the same number of non-transparent pixels (the original number of pixels per scallop as 41,193, 44,817, 42,869, 33,272, 42199, with a standard deviation of 4,449 pixels; after resizing the stimuli so that they approximately matched the average number of pixels per scallop, the pixels per scallop were 40,828, 40,791, 40,904, 40,796, 40,642, with a standard deviation on 95 pixels). The scallop image dimensions were set to 50 × 50 pixels in the actual study. The plate images were set to 250 × 250 pixels.

The exact scallop images used in each dish were randomly determined, as were their set positions on the plate (care was taken so that the scallops were placed and spaced apart to resemble the vertical and polygonal arrangements that had been used in Experiment 1). The experiment was conducted on the Internet using the Adobe Flash based version of Xperiment (http://www.xperiment.mobi).

### Design and procedure

The design was similar to that of Experiment 1 in that two plates of food were shown to participants on each trial, and the task was to decide on the plate that the participant most wanted to eat. Here, however participants undertook all 8 of the experimental trials, which differed in terms of the size of the plate shown (either both plates were large or regular sized), the shape of the plate (either both were square or circular), the arrangement of the food (either both were vertical or polygonal) and the number of food items (one plate there were 3 items, whilst there were 4 items on the other plate).

## Results

A log-linear analysis was performed, using Plate Size (regular, large) × Plate Shape (circular, square) × food Arrangement (vertical, polygonal) × food Items (3, 4) as the variables (the final model's likelihood ratio was $\chi^2(10) = 3.54, p = .99$). Only the Arrangement × Items $\chi^2(1) = 5.41, p = .021$ interaction was kept in the model. Separate Exact Binomial tests found that 4 items were preferred for vertically arranged items ($p < .001$; with 307 picking the 4-item dish and 93 picking the 3-item dish; 95% CI [19.20%–27.70%]) and for those arranged as a polygon ($p < .001$; 333, [13.22%, 20.78%]).

## Discussion

There was no statistically significant evidence to support the scenario that plate overcrowding influenced dish selection here. At first glance, the results of Experiment 2 are rather different from those of the preceding study. Here, by far the majority of our participants preferred the 4-item dishes. In Experiment 1, though, the magnitude of this
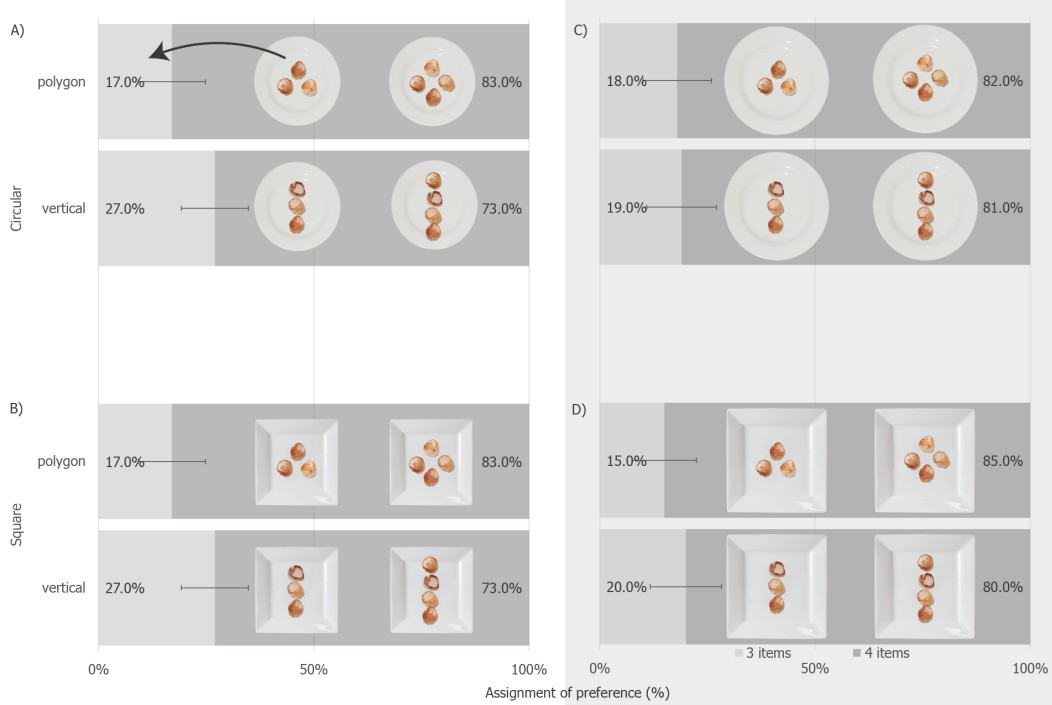

**Figure 4** The percentage of participants preferring one dish over the over for each of the Plate Arrangement, Plate Size, Food Shape and Food Item cells in Experiment 2 (error bars are 95% CI, all differences $p < .001$).

preference was much smaller; indeed, when the items were arranged vertically, participants preferred the 3-item dish over the 4-item dish. It should be noted, though, how the pattern of results in Figs. 4A and 3B, which tested participants on the same plate sizes as Experiment 1, if one ignores the magnitude of the preference difference, resembles that seen in Figs. 3A and 2B for Experiment 1: when the items were arranged vertically, more participants picked the 3 item dish, relative to when the items were arranged as a polygon. Given how Experiment 1 provided evidence of food arrangement and this study does not, we decided to continue exploring food arrangement in our subsequent studies. One possibility that we came up with what that we might just be looking at a ceiling effect here, and this might have led to this difference between studies.

Why do we observe such a discrepancy between this study and the previous, in terms of *magnitude of preference difference*? One possibility is that the population from which the participants were sampled are quite different to each other, with those in Experiment 1 predominantly coming from the UK (and of the sort who visit science museums), whilst those in this study mostly came from North America; indeed, a potential major driver here could be that North Americans generally have larger meal sizes (as explored in the movie 'Super Size Me', *Spurlock, 2004*).

# EXPERIMENT 3: EQUATING PORTION SIZES

In this experiment, we scaled the 4-item dish so that it contained exactly the same amount of food as the 3-item dish, by factoring in the height of the scallops. By so doing, we factor out the influence of portion size in this study (if we ignore the fact that perceived portion size often differs from actual portion size—as discussed in the 'Introduction'), which should give us a clear indication whether or not participants prefer one dish over the other for perceived portion size, or for the likely aesthetic difference between 3 or 4 elements on the dish. Once again, our hypothesis was that participants would prefer the 3-item over the 4-item dish. Note that plate size has been shown to influence perceived portion size (for this and other such influences, see *Benton, 2015*; *Hollands et al., 2015*). However, as we never contrast portions over differently-sized dishes, such effects should not confound the results reported here.

## Materials and methods

### Participants

One hundred (31 female) were recruited from Amazon's Mechanical Turk to take part in the experiment in return for a payment of .35 US dollars. The participants ranged in age from 18 to 69 years ($M = 33.1$ years, SD $= 10.9$). The experiment was conducted on 10/06/2015, from 16:00 GMT onwards, and over a one-hour period. The participants took an average of 89 s (SD $= 104$) to complete the study.

### Stimuli, design and procedure

This study was identical to Experiment 2 except that the scallops were scaled so that each plate contained the same amount of food. In the previous studies, the scallops were held within $50 \times 50$ pixel boxes, and we assumed that the height that the scallops were off the plate was approximately 2/3 of this measure (33.3 pixels). Thus, on a three-item plate, the scallops were each tightly held within a 250 000 voxel box ($3 * 50 * 50 * 33.33$). The scallops in the four-item plate were scaled along the $x, y$, and $z$ axes to 90.86% of their original size so that the boxes they were enclosed within also summed up to this value ($4 \times 45.43 \times 45.43 \times 30.29$).

## Results and discussion

A log-linear analysis, as defined in Experiment 2, was conducted using data from this study (the final model's likelihood ratio was $\chi^2(14) = 5.23$, $p = .98$). As in the previous study, the model only retained the effect of Items $\chi^2(1) = 41.77$, $p < .001$. 4-item dishes (selected 491 times, or 61.38% of the time) were 1.59 times more likely to be preferred more than dishes with 3 items (309 times; Fisher's exact $t$-test 95% CI were 57.90% and 64.76%).

The results indicate that, in actual fact, the 4-item dishes were preferred over the 3-item dishes. This result certainly runs contrary to the widespread claim that that odd-number of items are preferable. Unfortunately, however, a further confound may have swayed this result. Next, we tested whether our arrangements were thought different in portion size due to potential distortions brought about by psychological illusions of volume perception.

# EXPERIMENT 4: SCALING STUDY

## Materials and methods

### Participants

One hundred participants (51 female) were recruited from Amazon's Mechanical Turk to take part in this study in return for a payment of 1 US dollar. The participants ranged in age from 19 to 56 years ($M = 30.2$ years, SD = 8.02). The experiment was conducted on 5/06/2014, from 14:00 GMT onwards, over a period of three-hours. The participants took an average of 378 s (SD = 138) to complete the study.

### Stimuli

The individual scallops used in Experiment 2 onwards were dynamically sized, positioned and combined as a dish stimulus as required on each trial (on a plate in most trials; n.b., the plates used were those defined in Experiment 2). The exact scallop images used in each dish that were to be scaled (henceforth termed the 'scaling-dish') were selected randomly, as were their set positions on the plate (care was taken so that the scallops were placed and spaced apart to resemble the vertical and polygonal arrangements that had been used in Experiment 1). The scallops in each dish were simultaneously scaled using the scroll button on the mouse or the left and right cursor keys (where a 'toward the body' scroll and the left cursor key scaled the image downwards) – importantly, the distance between the centre points of the targets did not change on scaling. The minimum size scallops were scaled so that they tightly fit within a $25 \times 25$ pixel box. The maximum size was $150 \times 150$ pixels. The starting size of the scallops was randomly determined but was always such that the scallops fit within a box larger or equal to $40 \times 40$ pixels and smaller or equal to $60 \times 60$ pixels.

A target stimulus that was randomly selected from the 5 scallop stimuli was also present on each trial. This stimulus was always sized so that it fit within an $87 \times 87$ pixel box.

### Apparatus

The apparatus varied by participant as the experiment was conducted online. The experiment utilized 'full screen' mode (i.e., utilizing the entirety of the participant's monitor), and took place within a $1024 \times 768$-pixel box in the centre of the screen, irrespective of the size of the monitor.

### Design

A within-participants experimental design was used with all of the participants undertaking all of the experimental trials (trial order was randomised). The dependent variable was the computed scaling factor which the participant applied to the dish of scallops so that they would, together, match the volume of the Target stimulus. A scaling factor of 1 would indicate that the participants scaled the portion so it exactly matched the volume of the target, whilst values smaller than 1 indicate the scallops were sized such that they were of a lesser overall volume than the target. Independent variables were the size of the plate (large or regular), the shape of the plate (circular or rectangular), the number of scallops (three or four) and the arrangement of the scallops (vertical or polygonal). Further trials

Please use the scroll button on your mouse, or your left/right keys, to resize all the items in **portion 2** so that they are the same amount of food as **portion 1**.

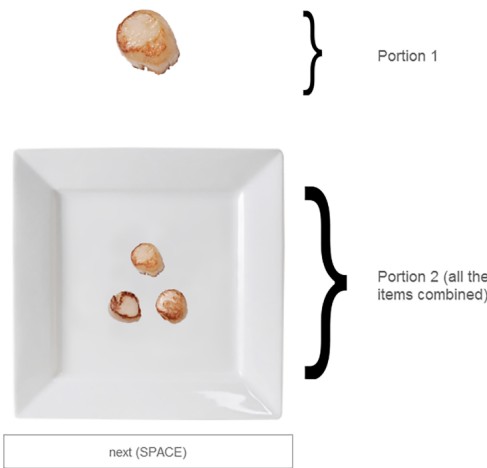

**Figure 5** The trial layout, as presented to participants in Experiment 4.

included dish variants where there were one or two scallops only (the latter, arranged vertically or horizontally) and where there was no plate present. Note that there were several further conditions, the data from which will be reported elsewhere (Woods et al., 2015, unpublished data).

### Procedure

On each trial, a screen as shown in Fig. 5 was presented. Participants had to scale the size of the scallops shown in the portion of the screen labelled Portion 2, so that they matched the same amount of food as shown in Portion 1. Although Portion 1 was the same size on all trials, the Scallop that was shown as Portion 2 randomly varied in default size across trials. There were a total of 35 trials. At the end of the study, we explicitly asked participants "When you did the task, were you resizing Portion 2 so that…," and offered two choice options "one food item was the same size as Portion 1," "all the items together in Portion 2 added up to the same amount as in Portion 1." The 20 participants who chose the first option were excluded from the analysis. There were 35 experimental trials, the data from 20 being reported here.

### Results and discussion

Eleven out of twenty sets comprising the data were not normally distributed $D(80)$, $p < .05$. Log transforming the data mostly corrected this issue, with only one set remaining non-normal, $D(80) = .94$, $p < .001$ (large round plates containing 3 polygonally-arranged items). The same set was also was significantly skewed, $p < .001$, and affected by kurtosis, $p < .01$. Another set was also affected by kurtosis, $p < .01$ (regular-sized round plates with 4 vertical items). 0.5% of the scaling data from each dish was found to be outlying (defined

as being larger or smaller than the mean +-3 standard deviations) and so was corrected (replaced with the nearest non-outlying data value, mean +-3 standard deviations).

With the majority of the cells of data now being normally distributed, one-sample $t$-tests were used to test whether the log-scores different from the null hypothesis of that no scaling was required, or log(1), with a Bonferroni corrected alpha threshold set to .05/35 (a further 15 tests on data not reported here were conducted in Woods et al., 2015, unpublished data). Only data for large round plates with 4 vertical scallops differed significantly $t(79) = 3.64$, $p < .001$, requiring scaling of 1.10 to be seen as the same size as the target food. As all other 4-scallop dishes did not so differ (as would be expected given the shift in 3 vs. 4 item preference seen in previous studies), we must assume the null-hypothesis that portion size distortions cannot really account for earlier findings (that 4-item portions were often preferred over 3-item portions).

Note, though, that in previous research the participants had to choose between 2 dishes, each of which could be differently influenced by scaling factors. Thus, potentially subtler distortions of size (not detectable when contrasting from baseline as done in the above tests that were essentially *between-participant*), between each pair of dishes, may have driven the shift towards the 4-item dish as opposed to 3-item dish from past studies. To explore this, a 4-way repeated measures ANOVA was conducted on the log scaling data with plate Shape, plate Size, Items and food Arrangement as factors. Items and Arrangement interacted $F(1, 79) = 22.86$, $p < .001$, $\eta_p^2 = .22$, with a post-hoc stepwise Newman-Keuls analysis (critical $p < .05$) showing that 4-scallops arranged as a polygon requiring more scaling (mean 1.04) than the other conditions (.97; significant main effects that were involved in these interactions are not reported). What this means, in fact, is that the 4-scallop polygon arrangements required were seen as the *smaller* portion than on other dishes (it was required to be scaled by a factor of 1.04, whilst the other dishes had to be scaled by .97, to both be seen as *the same size* as the target portion). We would have expected it to be seen as bigger than the other dishes, for it to explain the apparent 4 item preference over 3 items as seen previously.

Recall the pattern of results from Experiments 1 and 2, where the preference ratio of 3-item polygonal scallops to 4-item polygonal scallops was greater or more severe than that for vertically arranged items. The fact that here, 4-items are perceived as a smaller portion than 3 items may be linked to this pattern, although at this moment in time, it is unclear how.

Several other distortions, albeit smaller in magnitude, were also found. Shape and Size also interacted $F(1, 79) = 5.85$, $p < .018$, $\eta_p^2 = .07$, with the same post-hoc procedure revealing that large-round plates required its contents to be scaled more to match the target (mean 1.02) as compared to regular-round (.97), large-square (.99) and regular-square plates (.98). Large-square plates required more scaling than round-regularly sized plates.

A separate repeated measures ANOVA was used to test whether the trials in which there were no plates (only scallops were shown) differed in terms of scaling required to match the target stimulus. Items and Arrangement were used as factors. There was a main effect of Items, $F(1, 79) = 8.47$, $p = .01$, $\eta_p^2 = .10$, with the 4-item displays (mean 1.04) requiring significantly more scaling than the 3-item displays (1.00).

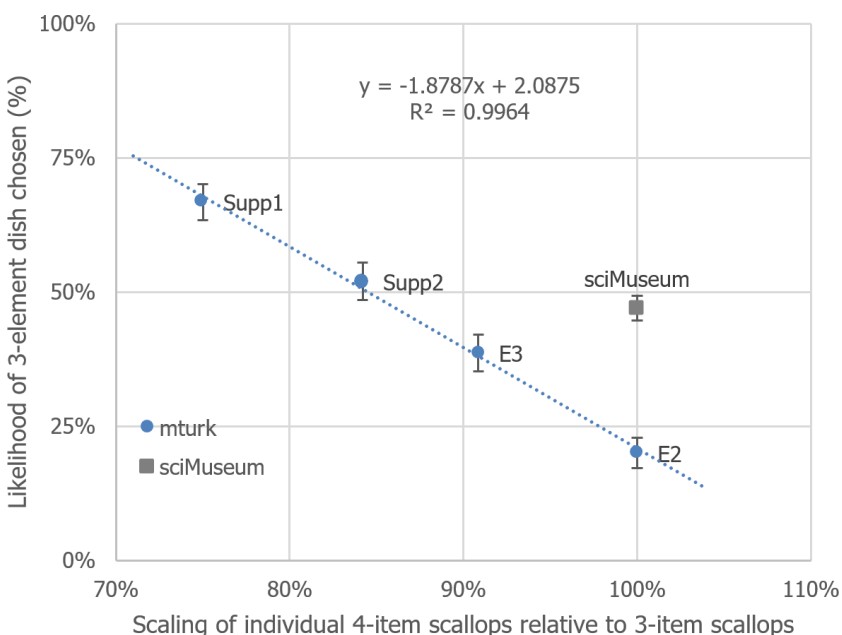

**Figure 6** **Depiction of the relationship between 4-item scallop scaling and likelihood of 3-scallop dish chosen, over the experiments reported so far.** Error bars represent the 95% CI derived from separate Fisher's exact binomial tests.

## Discussion

Although some stimuli from previous experiments were indeed affected by size distortions, there was no systematic effect of distortion of 4-item dishes to appear *larger* than the 3-item dishes, the result of which could be leading participants to prefer 4-items over 3.

The tentative conclusion that could now be drawn is that the even number of items on a plate are preferred over odd numbers of items. To say so, though, one must ignore several important issues highlighted in the introduction, such as whether 3 vs. 4 items generalise to odd vs. even number of items, as well as whether the effects observed here are only applicable to our scallop stimuli.

## COMBINED ANALYSES

The preceding experiments have highlighted the importance of perceived portion size on dish choice, with larger portions tending to be selected over smaller portions. This relationship has been quantified in Fig. 6 as a simple-regression model, which shows an extraordinary linear relationship between these factors, for all studies, except the very first one.

So, the question remains as to why the results from the Science Museum study differ so much from the data collected from Mechanical Turk for Experiments 2–5? Recall that the scallops in our original study were not yet scaled to be equal in size in terms of pixels, as done from Experiment 2 onwards. Could the 'fixed' stimuli used in Experiment 1 have led to the above discrepancy? To test for this, we isolated each dish in the study using

**Table 1** Detailing the size, in pixels, of each scallop that was used in Experiment 1.

| Plate shape | Food shape | Food items | Pixels | Pixels per scallop |
|---|---|---|---|---|
| Circle | Polygon | 3 | 4,606 | 1,711 |
| Circle | Polygon | 4 | 6,612 | 1,702 |
| Circle | Vertical | 3 | 5,107 | 1,650 |
| Circle | Vertical | 4 | 5,919 | 1,653 |
| Square | Polygon | 3 | 4,545 | 1,535 |
| Square | Polygon | 4 | 6,427 | 1,607 |
| Square | Vertical | 3 | 5,133 | 1,480 |
| Square | Vertical | 4 | 6,598 | 1,515 |

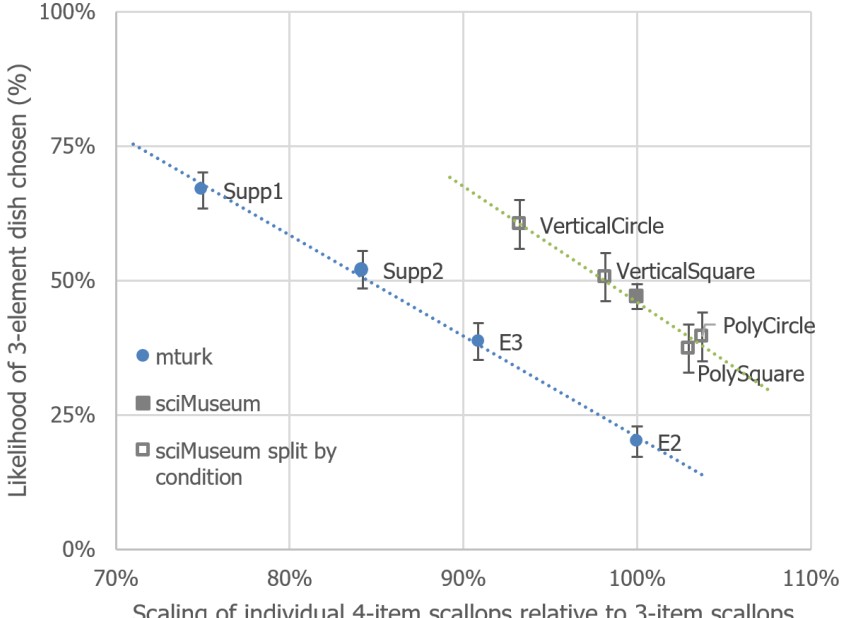

**Figure 7** Identical to Fig. 6, except that the conditions from Experiment 1 have been added individually as transparent black bordered squares.

photo-editing software to estimate total scallop pixels (see Table 1). We then calculated the individual scaling factor present for each condition (square plate × vertical items, .98; square × polygon 1.03; circular × vertical, .93; circular × polygon, 1.04) and plotted this on Fig. 7, alongside the ratio of 3 items being selected for each condition.

Although with 4 data points per model, any inference from statistical analysis must be treated with caution, the updated MTurk model's gradient (−.53, 95% CI [−.63− −.43]) and Science Museum model's gradient (−.45, [−.67−−.24]) are similar; it is their $y$-axis intercepts that potentially differ (Experiment 2–5, 111%, [106%, 116%]; Science Museum, 121%, [110%–131%]; n.b. overlapping CI).

Why would there be this upward shift of preferring 3 items as opposed to 4 in the Science Museum study? After further careful investigation we discovered that the images that had been used in the Science Museum study had been arbitrarily scaled by the designers

of the citizen science platform that they had been presented on so that they were 67.1 % smaller in width and height than their original file size (images contained within a 380 pixel × 255 pixel image-file). Furthermore, it also transpired that there were *two* sizes of the original stimuli, with the original images we used being both smaller in scale and held within a differently dimensioned image-file (372 pixels × 306 pixels). By using graphical editing software, we were able to estimate that the Science Museum images were 85.2% smaller in width and height to the images used in Experiments 2–5. Could the difference in overall food size lead to this apparent upward shift between models as seen in Fig. 7? We test this hypothesis next. We also tested whether participants' hunger influenced their dish choice.

## EXPERIMENT 5: DIFFERENCE DUE TO OVERALL SIZE OF STIMULI?

In this study, participants undertook a version of the task reported previously where we systematically varied the physical sizes of the dish stimuli on the screen. Both 3- and 4-portion stimuli were resized to the same degree. Note that the monitors of our online participants and thus the stimuli presented differ in terms of size across individuals. To get around this issue, we used a repeated measures design such that all of the participants undertook the trials where differently sized stimuli were presented.

It was hypothesised that if the variation in the size of the stimuli was indeed responsible for the difference between the Science Museum study and all of the other studies (as shown in Fig. 7), in this study, we should observe a shift in dish preference as we scale the stimuli from smaller to larger in size from that observed for the Science Museum study to that observed for the Mechanical Turk experiments.

We also tested whether the participant's self-reported hunger level influenced the choice design in this task by asking participants how hungry they were.

### Materials and methods

One hundred participants (40 female) were recruited from Amazon's Mechanical Turk to take part in the experiment in return for a payment of .50 US dollars. The participants ranged in age from 20 to 67 years ($M = 34.8$ years, SD $= 11.2$). The experiment was conducted on 15/06/2015, from 14:30 GMT onwards, and over a 45-minute period. The participants took an average of 105 s (SD $= 58$) to complete the study.

### *Stimuli, apparatus*

The stimuli were the same as reported in Experiment 1, except that the scaling of both the 3-item and 4-item dishes (as well as plates) were varied, relative to the original size of the 3-item stimuli as used in Experiment 2. We decided to size the stimuli at 100% of those used in Experiment 2 (50 pixels along one dimension), same size of the Science Museum study (42.6 pixels; a difference of 7.39 pixels), smaller than the Science Museum by 7.39 pixels, and larger than the one used in Experiment 2 by 7.39 pixels. In order of size, the stimuli were scaled to 70.44%, 85.22%, 100% and 134.28% of the stimuli used in Experiment 2 and onwards (henceforth termed Small, SciMuseum, Regular, Large).

### Design

We used a fully factorial design here with all participants completing all of the experimental trials. The design was identical to that in Experiment 1, except that an additional factor of plate Size (regular versus large) was included. We also had the participants report their hunger level.

### Procedure

The procedure was identical to that used in the previous studies except that we also assessed participants' self-reported hunger by means of scales anchored on the left hand side with "not hungry at all" and on the right "very hungry." Hunger scores from this scale varied from 0 to 100.

## Results and discussion

A log-linear analysis was performed, as defined in Experiment 2, but with the additional independent variable of plate Size, using the data from this study (the final model's likelihood ratio was $\chi^2(30) = 5.12$, $p = 1$. The only factor to be retained by the model was Items, $\chi^2(1) = 138.91$, $p < .001$, with 4-item dishes (selected 1034 times) 1.83 times more likely to be chosen than 3-item dishes (selected 566 times). The Exact Binomial test 95% confidence intervals for this effect (33.03%, 37.78%) intersected the value predicted by the model for a scaling of 90.86% for the 4-item scallops (37.62%). The lack of any effect of Size means that the Small (3-items chosen 33.50% of the time, 95% CI [28.89%–38.36%]), SciMuseum (35.00%, 30.33%, 39.90%), Regular (36.75%, 32.01%, 41.68%) and Large sizes (36.25%, 31.53%, 41.17%) did not significantly differ from each other in terms of the ratio of participants who chose 3-item versus 4-item dishes.

To test whether the hunger level of the participant influenced their dish choice, a correlation analysis was used to test for a relationship between the total number of times each participant chose the 4-item dish, and their self-reported hunger score. As the 4-item dish was 1.83 times more likely to be chosen than the 3-item dish (as reported above), we would then expect that, if hunger was an important factor, participants who were more hungry would be more likely to choose the 4-item larger in portion size dish; this was not the case, $r = -.12$, $n = 100$, $p = .25$.

There was no evidence that the difference in size between stimuli used in Experiments 2–5 and those used in Experiment 1 was responsible for the difference in $y$-axis intercept. There are several possible reasons for this. One possibility is that the within-participants design of this study could have prevented any effects being detectable. For example, consider that the participants here saw many trials one after the other, involving the same task, "Which dish do you prefer?" Potentially, after undergoing several such trials, the participants may have 'made up their mind' as to how to respond to each trial (e.g., "I like big portions, so I will always pick the larger portion"), which could sufficiently dilute any normally detectable effects so that they became undetectable. In the Science Museum task, however, the maximum number of trials undertaken by the participants were 2, with the majority of trials thus requiring cognitive effort rather than relying on a quick heuristic.

Another possibility is that the populations from which participants from Experiments 2–6 were sampled from differed in some key criteria from those who undertook the Science Museum experiment. We test this in the next section.

## EXPERIMENT 6: DO THE EFFECTS HOLD WHEN SAMPLING FROM A DIFFERENT POPULATION?

A logical step is to rerun the study, but with a different group of participants. Psychology students are well known for being WEIRD (Western, Educated, Industrialised, Rich, and Democratic individuals; see *Henrich, Heine & Norenzayan, 2010*) and different from Mechanical Turkers (discussed in *Woods et al., 2015*). Here, we recruited participants from the up-and-coming cloud-sourcing platform Prolific Academic, which actively recruits participants with no geographic criteria for potential participants (although if desired, an extensive range of filters can be used to define the subpopulation of participants one needs for a given study), as opposed to MTurk, whose participants are typically North American.

If the difference between the data from the Science Museum reported in Experiment 1, and the rest of the studies reported so far is indeed attributable to some difference over populations, Prolific Academic participants may differ from both these groups too.

To test whether this is so, we collected data from stimuli that are sized according to those reported in Experiment 2, 3, and Supplemental Information. We should observe the same gradient as found previously, but with a shift in the $y$-axis intercept.

### Materials and methods
#### *Participants*
391 participants (162 female) were recruited from Prolific Academic to take part in this study in return for a payment of .35 US dollars. The participants ranged in age from 18 to 67 years ($M = 28.4$ years, SD $= 9.1$). 142 participants reported being from the United States, 135 from the United Kingdom, 17 from India, 13 from Canada and 5 from Portugal (country frequencies of fewer than 5 individuals are not reported). The experiment was conducted on 3/07/2015, from 16:00 GMT onwards, over a period of six-hours. The participants took an average of 106 s (SD $= 48$) to complete the study.

#### *Stimuli, design, procedure*
Identical to Experiment 2, except that Large Plate condition was removed and an additional between participant factor of Scaling was included (how large the 4-item stimuli were, relative to the 3-item stimuli, the levels being 100%, 91%, 84%, and 75%).

### Results and discussion
A log-linear analysis was run using data from this study (the final model's likelihood ratio was $\chi^2(14) = 3.80$, $p = 1$. Items × Scaled × Arrangement interacted $\chi^2(3) = 10.54$, $p < .02$. Eight separate Bonferroni corrected Fisher's Exact tests were used to explore this interaction, the results of which are detailed in Table 2.

The Items × PlateShape interaction was also significant, $\chi^2(1) = 4.34$, $p < .05$ with follow-up Exact Fisher tests for each Plate Shape revealing that Square Plates with 4 items

**Table 2** The percentage preference for 3-items relative to 4-items in Experiment 6 (as derived from Bonferroni-corrected exact Fisher's tests; 95 % CI in brackets).

| | Scaling of 4-item scallops to 3-item scallops | | | |
| --- | --- | --- | --- | --- |
| | **100%** | **91%** | **84%** | **75%** |
| Polygonal arrangement | 30.39[***] (24.16, 37.20) | 25.27[***] (19.20, 32.15) | 44.33 (37.22,51.62) | 62.24[**] (55.06, 69.06) |
| Vertical arrangement | 47.55 (40.53, 54.64) | 45.70 (38.39, 53.15) | 49.48 (42.25, 56.74) | 64.29[***] (57.15, 70.99) |
| n | 204 | 186 | 194 | 196 |

[**]p<.01.
[***]p<.001.

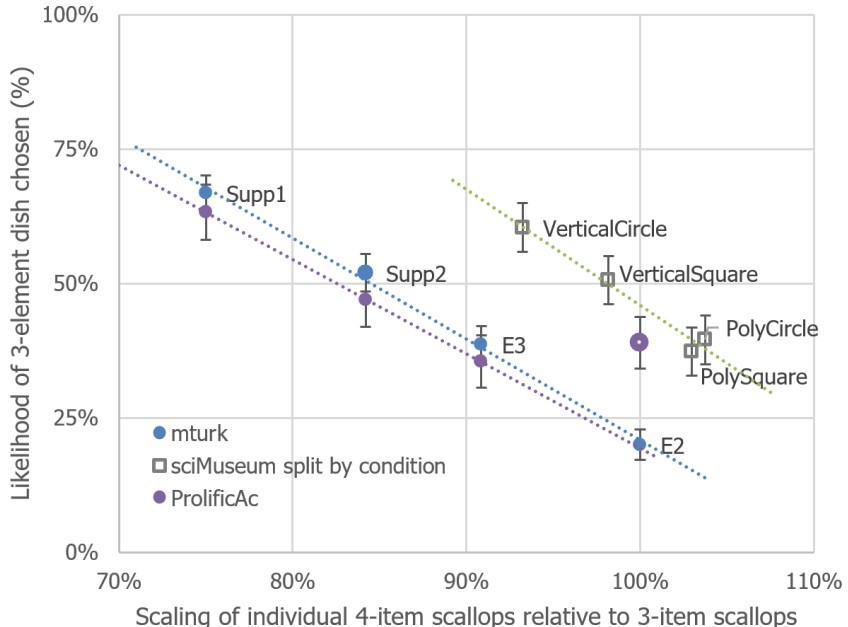

**Figure 8** **Identical to Fig. 7 but with the results of Experiment 6 added.** Note that the large transparent purple point did not follow the pattern of the other data points from this study.

(selected 440 times) were selected 1.29 times more frequently than Square Plates with 3 items $p < .001$ (selected 340 times). There was no such difference for Round Plates (3 items selected 381 times, and 4-items 399 times).

The data for this experiment has been plotted alongside the previous experiments in Fig. 8. Whilst the scaled data points for 91%, 84% and 75% form a straight line that does not appear to differ from that of the past MTurk experiments (gradient, −.57, 95% CI [−.64−−.5]; intercept 111%, 95% CI [107.53%–114.56%]), the data from the 100% scaled condition unexpectedly does not fit this profile (the thick transparent purple point in the figure).

Back in Experiment 2 (labelled E2 in Fig. 8, in the lower right quadrant along the $X$-axis 100% mark) we observed an Arrangement × Items interaction and speculated that this arose due to overcrowding on the plate for the 4-item in relation to the 3-item vertical dishes. One possibility in the current study is that the vertically aligned scallops were
likewise seen as overcrowding the plate. For some reason, however, the participants here *preferred* this compared to when the items were not so overcrowded, hence the 4-item preference from 3-item preference shifted upward, as shown in the graph.

Unfortunately, due to the confound of stimuli sizing for the Science Museum study discussed in the 'Combined Analysis' section, we do not have data for Vertical dishes at this level of 4-item scaling. We do, however, speculate that such an effect would be present, and would increasingly influence the results as overcrowding increased yet further. As overcrowding is not, however, the focus of the present research, we will leave the speculations of the drivers of this finding to future research.

In terms of our initial hypothesis, despite the above unexpected finding, there is little evidence to support the idea differences in terms of population led to the shift in intercept between MTurk studies reported here, and the results of the Science Museum. In the General Discussion, we flesh out reasons why this may be the case.

## EXPERIMENT 7: 1–6 SCALLOPS PER PLATE

Our reviewers rightly pointed out a potential additional confound that we had missed, that of 'numerosity.' In all of the past studies that have been reported so far, participants were shown dishes where the one with an odd number of items *always* contained *fewer* items than the dish containing an even number of items. It could therefore be that, rather than participants referring an even number of items, they simply preferred the plate with more food items on it, as has been previously observed in both animals and human infants (see *Hauser, Carey & Hauser, 2000*; *Uller & Lewis, 2009*).

Here, we test a range of plate pairs containing one to six elements, with some plate pairs (all of which whose food was scaled, as done previously, to appear to contain the same amount of food) where the odd numbered plate has fewer elements than the even numbered plate, and where the even numbered plate has more than the odd number. We also test pairs of plates that both contain differing numbers of odd items, and even items. No variation in choice would be expected over these plate-pair configurations if it is numerosity that drives the previously observed effects.

### Materials and methods

One hundred participants (44 female) were recruited through Prolific Academic to take part in the experiment in return for a payment of .45 UK pounds. The participants ranged in reported age from 18 to 60 years ($M = 33.7$ years, SD $= 11.9$). The experiment was conducted on 8/11/2015, from 10:00 GMT onwards, and over a 3-hour period. The participants took an average of 169 s (SD $= 61$) to complete the study.

### *Stimuli, apparatus*

The scallop images were the same as reported in the previous study. These were combined into stimuli of 1–6 scallops, with the individual scallop-images scaled so that the stimuli contained the same amount of food (using the procedure incorporating depth, as detailed in Experiment 3; the scaling factor used for $x$, $y$, $z$ dimensions of 1, 2, 3, 4, 5, 6 scallop stimuli were respectively 1.44, 1.14, 1, 0.91, 0.84, 0.79). The initial scallop-image was

placed centrally. Subsequent image positions were determined by randomly selecting one of the already placed images and then generating a random point 55 to 82.5 pixels from its centre. If the new position was within 55 pixels of any already placed image, the process was repeated. If this was unsuccessful after 10,000 attempts (this never occurred during development, with typically <100 attempts needed per placement), the point of farthest distance from existing images so far generated was used. This led to stimuli consisting of scallop-images that were clustered around a central point. When needed, a sixth scallop-image was chosen at random from the 5 original scallop images. The pair of stimuli shown on each trial differed in terms of the number of scallop-images, with this number always being by 1 or 2 scallops (the pairs consisted of: 6 vs. 5 scallops, 6 vs. 4, 5 vs. 4, 5 vs. 3, 4 vs. 3, 4 vs. 2, 3 vs. 2, 3 vs. 1, 2 vs. 1).

### Design

We used a near fully factorial design here with all participants completing all of the experimental trials. The independent variables were the number of scallops on each plate (ranging from 1–6) and the difference in this number of scallops over plates in a stimulus pair (1 or 2; n.b. we could not test 2- vs. 0-items, thus our design is incomplete). The dependent variables were the dish out of each pair that was preferred, and, as a control, the dish that the participants thought contained the most food.

### Procedure

The procedure was identical to that used in the previous study. As before, participants had to select which of two stimuli they preferred (there were 9 such trials). After, the participants were shown the same pairs of stimuli (that were *identical* in terms of individual scallop position) and asked for each, which dish contained the most food. There were 18 trials in total.

### Results and discussion

To test whether the groups differed in terms of whether the dish with the most items was selected in preference over the dish with a fewer number of items, all 9 groups were entered into a log-linear analysis under the factor of Group, along with this Most Preferred factor (see Table 3). The model, whose likelihood ratio was $\chi^2(16) = 14.71$, $p = .55$, only retained the Most Preferred factor $\chi^2(1) = 53.33$, $p = .001$, with participants 1.63 times as likely to choose the dish that contained more scallops as opposed to the plate with fewer scallops. The analysis was repeated with the dependent variable of Most Amount (whether the participant had chosen the dish with the most items as containing more food, over the dish with the fewer items) instead of the Most Preferred factor (the likelihood ratio was $\chi^2(16) = 11.52$, $p = .77$). The factor of Most Amount was (barely) retained $\chi^2(1) = 4.00$, $p = .04$, with dishes with fewer items thought as containing *more* food 1.14 times as often than dishes with more items. The result could indicate that participants preferred the dishes with seemingly *less* food on them. If this were so, we should expect that the difference in the perceived amount of food to tally with the degree to which a plates differed in preference; with our limited sample, we conclude, be it very tentatively, that there was no evidence for this, $r = .27$, $n = 9$, $p = .48$.

**Table 3  The percentage preference for the different dish pairs in Experiment 7, sorted according to preference magnitude.** Note how the column defining whether the dishes were odd or even seems relatively random in terms of order, which implies no relationship between this and strength of Preference.

| Items | | | Dish1, Dish2, odd or even items | Dish with more elements, by % of participants, is | |
|---|---|---|---|---|---|
| Dish1 | Dish2 | Difference | | Preferred (95% CI) | Bigger portion (95% CI) |
| 1 | 3 | 2 | OO | 73 (63.2, 81.39) *** | 47 (36.94, 57.24) |
| 2 | 4 | 2 | EE | 70 (60.02, 78.76) ** | 40 (30.33, 50.28) |
| 4 | 5 | 1 | EO | 66 (55.85, 75.18) * | 42 (47.71, 67.8) |
| 2 | 3 | 1 | EO | 62 (51.75, 71.52) | 42 (32.2, 52.29) |
| 1 | 2 | 1 | OE | 62 (51.75, 71.52) | 44 (34.08, 54.28) |
| 3 | 5 | 1 | OO | 59 (48.71, 68.74) | 50 (39.83, 60.17) |
| 5 | 6 | 2 | OE | 57 (46.71, 66.86) | 48 (37.9, 58.22) |
| 4 | 6 | 2 | EE | 57 (46.71, 66.86) | 60 (30.33, 50.28) |
| 3 | 4 | 1 | OE | 53 (42.76, 63.06) | 47 (36.94, 57.24) |

*$p < .05$, as derived from Bonferroni-corrected exact Fisher's tests; 95% CI in brackets.
**$p < .01$.
***$p < .001$.

Another consideration is that hungry people typically prefer larger portion sizers (e.g. *Brunstrom et al., 2008*), so if our effect of smaller portions being seen as larger did hold sway in people's dish preference, one would expect hungrier individuals to be more so influenced. However, when we re-ran the original log-linear analysis but including an additional variable of median split (which was 52.02 on our hunger scale; 1st and 3rd quartiles were 23.29 and 72.12), this factor was not included in the final model.[5]

We also conducted an analysis in which the number of items on the more numerous plate (3, 4, 5, 6), as well as the difference in the number of items between plates (1 or 2) were entered as independent variables into a log-linear analysis, alongside the Most Preferred factor as defined previously (the likelihood ratio of this model was $\chi^2(14) = 14.71$, $p = .40$; note that the 1 vs. 2 condition had to be excluded from this analysis to avoid empty cells). As before, the only factor retained by the model was again Most Preferred, $\chi^2(1) = 47.52$, $p < .001$).

So, there is evidence here that dishes with *more* items on them are preferred to items with fewer items. This is despite two key points. The first is that the portions on each plate were approximately equated in terms of food (in actual fact, there was limited evidence for the fewer itemed dish seeming to contain more food than its more numerous counterpart stimulus, which is in line with the results of Experiment 4), which is bolstered by the fact that hunger did not influence preferences. The second point to note is that the larger differences in the number of items did not lead to more exaggerated preferences for the more numerous dish.

If one considers that the dependent variable here represents the *number* of individuals who have a preference for one stimulus over another, and that we have controlled for effects such as the volume of food which would likely influence participants' decisions if they were hungry, our results could simply reflect individual preference (some individuals

[5]The model, whose likelihood ratio was $\chi^2(34) = 22.33$, $p = .94$, again only retained the Most Preferred factor $\chi^2(1) = 53.33$, $p = .001$.

prefer plates with a lot of food elements, others prefer less elements), that is unchanging over the manipulations introduced in this study.

## GENERAL DISCUSSION

Taken together, the results of the 7 experiments reported here provide no support for the commonly-stated assertion that an odd number of items on a plate would be preferred to an even number. After controlling for portion size (Experiments 2–3, Supplemental Information), testing for plate overcrowding (Experiment 2) and perceptual distortions (Experiment 4), only one group of participants were found to sometimes prefer 3-item dishes as opposed to 4 (Fig. 3; Experiment 1, the Science Museum); in contrast, two further groups of participants recruited through MTurk (Experiments 2–5) and Prolific Academic (Experiment 6) preferred 4-item dishes over three. Indeed, in our final study (Experiment 7), we found that the plate with the more food items was generally preferred, over that containing fewer items. We will discuss several major issues with these findings after briefly summarising each of the experiments in turn.

### Overview of the studies

The results of the first experiment, conducted in collaboration with the Science Museum with 1,816 participants, were ambiguous, with 3 items being preferred over 4 items when those items were vertically orientated and on a circular plate only. In all most other conditions, 4 items were preferred. This was followed up with a series of experiments that, in turn, tested, and helped control for several confounds, the first of which was ensuring that the individual food items were the same size over conditions (not so in the first study).

Next, we tested whether plate overcrowding had influenced findings in the first study. Experiment 2 explored this potential confound by testing whether the ratio between plate size and the surface area covered by the food influenced the plating preference. There was no statistical evidence for such an effect, although, descriptively, effects of food liking were less strong on larger plates than on regular plates, which warrants future research. Unexpectedly, 4-item dishes were preferred in all experimental conditions.

Several further experiments tested whether the difference in portion size over conditions in Experiment 1 acted to confound the results. The relative size of the 4-item portion was reduced relative to the 3-item portion in Experiment 3, and in Supplemental Information, with the general finding being that the larger the portion, the more people were likely to pick that portion over a smaller portion.

Contrary to the commonly-held belief, 4-items were preferred over 3 when portion sizes were equated. In Experiment 4, we tested whether there was a perceptual distortion of portion sizes such that the 4-item dish seemed greater in size than the 3-item portion, but there was no real evidence for this. This issue is explored in a complementary paper (Woods et al., 2015, unpublished data).

By means of a combined analysis, there was clear evidence that portion size plays a key role in deciding which plating people prefer, with larger perceived portions more likely to be chosen. Furthermore, we found that quantifying the portion sizes over experimental condition, the Science Museum study, seemed to obey this principle as well. However,

whilst the rate of change of the findings over the first study versus other studies seemed equivalent (that is, portion size change tallied with liking change), the scaling at which a 4-item dish required to seem the same size as the 3-item dish differed.

One explanation for this variation was that all the stimuli used in Experiment 1 were actually smaller than those used in the subsequent studies. When we explicitly tested for this with a repeated-measures design in Experiment 5 (to get around the issue of hardware variation in online research), this issue was, however not found to influence plating preferences.

Another explanation was that population differences from which Experiment 1 participants were from (the general public in the UK mostly) and those recruited from in other studies (Mechanical Turk) led to this shift. Experiment 6 attempted to test this by recruiting from a third population (Prolific Academic) to see whether this population's preferences differed from the other two populations; these individuals though also adhered to the same portion size dish preference principle. This new sample did not really differ from the samples recruited through Mechanical Turk, but nevertheless we cannot rule out that population and/or cultural differences have indeed caused the discussed difference in results. Furthermore, it seemed that plate overcrowding has a different impact on plate preference for this group, than for Mechanical Turkers.

Finally, we demonstrated that it was the number of items on the plate, rather than whether the plate had an odd or even number of items that influenced which of two plates of food our participants preferred, with the majority preferring the plate containing more elements. As the dishes in this study contained approximately the same amount of food this could not be due to our participants deciding on the plate that would seem most filling. The sheer difference in the number of elements between plates also failed to influence preference ratings leading us to speculate that, in this particular study, there were simply more participants who generally preferred plates containing more elements than fewer elements. Presumably then, we would expect the same study ran with those recruited from Experiment 1 to reveal the opposite trend.

## Caveats
### Generalizability
Just how generalizable are the results obtained here with the scallop stimuli? Consider that larger scallops are typically more expensive and presumably preferred by both the chef and consumer, but which likely require more effect to cut up and eat. This trend, however, does not necessarily hold for other food types. One way of answering such a question of Generalizability would involve surveying just how frequently different numbers of various food items appears in natural dining situations (see *Michel et al., 2015*; *Michel, Velasco & Spence, 2015*, for a methodology that could help elucidate this mystery). Another consideration is whether these findings extend to general public. Is this only an issue with western chefs?

### Experimental design
Here, the pairs of dishes presented to participants were mostly identical in terms of plate shape, plate size and food arrangement (vertical versus polygonal), which meant that it was

impossible to test for interactions between these factors. The decision to go with this design was to minimise the number of trials participants would have to undertake (Experiment 2 onwards) or to ensure sufficient numbers of participants per group in the Science Museum study (we were delighted that 1,816 participants took part in our study and expected a smaller sample size).

## CONCLUSIONS

The evidence reported in this study suggests that a plate containing more items is generally preferred over a plate containing a smaller number of items, despite those plates containing the same amount of food. It seems likely though that such an effect varies over populations and cultures (cf. Experiment 6), though future research will be needed to verify this claim. To arrive at a clear result on this topic will be challenging, as several major issues pertaining to the experimental design and generalizability of the findings still need to be controlled for. Some tangential findings arose as a result of teasing apart the initially promising findings in the original study performed at the Science Museum. Although not significant for 3 or 4 items on a plate (Experiment 4), we report evidence in a complementary article that the perceived portion size of vertical and horizontal pairs of items is distorted (Woods et al., 2015, unpublished data). There was also suggestive evidence for the negative impact of plate overcrowding on liking, but, again, this warrants further studies to be verified and properly tested for.

## ACKNOWLEDGEMENTS

We are grateful for suggestions for explanations for the results of Experiment 1 by Moritz Bernoully, Virtyt Gacaferi, Ellen Jarvis, Jens Karraß, Johanna Kuenzel, Daniel Lakens, Alejandro Salgado-Montejo, Emily Snowden, Winni Theis, Catherine Transler, Pepijn Vemer, Kyra Woods, and Simon Woods. We would also like to thank the hundreds of Mechanical Turk and Prolific Academic participants who took part in our studies.

### Funding
CS received funding from the AHRC for the Rethinking the Senses project (AH/L007053/1). The funders had no role in study design, data collection and analysis, decision to publish, or preparation of the manuscript.

### Grant Disclosures
The following grant information was disclosed by the authors:
AHRC:AH/L007053/1.

### Competing Interests
Andy Woods is the founder and employee of Xperiment, Surrey, United Kingdom.

## Author Contributions

- Andy T. Woods conceived and designed the experiments, performed the experiments, analyzed the data, wrote the paper, prepared figures and/or tables, reviewed drafts of the paper.
- Charles Michel and Charles Spence conceived and designed the experiments, performed the experiments, wrote the paper, reviewed drafts of the paper.

## Human Ethics

The following information was supplied relating to ethical approvals (i.e., approving body and any reference numbers):

Oxford University's Medical Sciences Inter-Divisional Research Ethics Committee (approval # MSD-IDREC-C1-2015-004).

## Data Availability

Raw data is available in the Supplemental Information.

## Supplemental Information

Supplemental information for this article can be found online at http://dx.doi.org/10.7717/peerj.1526#supplemental-information.

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
