# Peer review of "Odd versus even: a scientific study of the ‘rules’ of plating"

_PeerJ, doi:10.7717/peerj.1526_

## Round 0.1 · original submission · Major Revisions

Dear Dr. Woods and coauthors,

Two reviewers have now commented on your manuscript. Both acknowledge that the research described in the ms is of good quality and original. I add that it is interesting, as it tackles the problem of assessing empirically whether a traditional say in kitchen/cuisine is driven by an actual preference in the food consumer. This is pretty much in line with the scientific approach towards assessing 'culinary precisions' and 'kitchen stories' in the visual domain underlying plating and composition, rather than in the reasonably richer tradition concerning the chemical senses.

However, both reviewers also believe (independently) that the experiments carried out in the study do not exclude that another potential factor, besides parity and surface/volume quantity, might have a determinant role in observers' choice, namely number of items. Both reviewers thus suggest to consider numerosity of items as a factor in a control experiment. Overall, the suggested conditions for further comparisons are 4 v 5 / 4 vs 6 / 2 vs 3. I am quite convinced that a control experiment in this direction would add clarity to the current results and their interpretation, so I am suggesting to include it in a revised version, also taking into account the other comments expressed by the reviewers.

I would also suggest to better substantiate the issue of 'odd better than even' in the Introduction, as regards the sources of this belief. It is unclear to me whether this is purely orally transmitted wisdom, as opposed to formal education in the field of cooking (thus possibly present in written sources, handbooks, etc). I am suggesting this check based on the personal impression that the issue might be not necessarily associated with visual hedonics (i.e. strengthening the liking of food in the observing consumer), but possibly with matters of etiquette. Wouldn't it be possible that, given the assumption that food on a dish might be consumed only partially (and this would very much apply to a context of cuisine rather than kitchen), an odd-item presentation would favor a division with remainders?

·

Basic reporting

No Comments

Experimental design

In general, experimental designs are quite good. However, the authors could have considered the variable "numerosity" along with "odd vs even food items". I'll try to explain my point. The authors find predominantly an effect that seems to confirm a preference for an even number of food items distributed on a dish rather than an odd number: with exception for one condition in exp. 1, participants seem to prefer dishes with 4 items instead of 3. Such preference does not seem to be driven by portion size of the single scallops or by crowding. However, there could be a confound here, because participants might just prefer dishes with more scallops. My hypothesis is that if the odd number was 5 and the even 4, or the odd number was 3 and the even 2, then dishes with the odd number of scallops would have been chosen over dishes with the even number. What I want to say is that along with all the controls the authors performed, it might have been worthwhile to run yet another experiment in which participants could chose between dish pairs made of 3items/4 items, 4 items/5 items, and of course 3 items/5 items. My bet is that data would have been much clearer, in that people may actually just be interested in portion size (i.e. size of the portion or number of items constituting a portion). Eventually, if my hypothesis holds, then the motivation behind such preference would be another argument worth investigating, as it does not seem to be driven by an aesthetic difference.
Finally, in conducting exp. 6 why didn't the authors just collect data from UK people? Wouldn't that be the perfect control for the "cultural" issue?

Validity of the findings

Only one comment: Along with acknowledging, as the authors correctly did, that dishes with an even number of food items are mostly preferred over dishes with an odd number, the authors should also ackwoledge that there may be at least two more competing hypothesis which still need to be addressed. First, as briefly discussed in the section above, the preference may not be driven by an intrinsic aesthetic value linked to even numbers vs odd numbers of items displayed on a dish, but on another factor that can be called as "numerosity" or "number of food items": within a range of small numbers (e.g. 1, 2, 3, 4, 5, 6, 7) dishes that display small food in higher number may be preferred over dishes that display the same kind of food but in a smaller number (the definition of small and big numbers may then also depend on food item size). Secondly, cultural factors cannot still be ruled out (see discussion of Exp. 6).

Additional comments

Just some small typos:
- line 46: "through to food" does not sound right.
- line 293: there is a double parenthesis .
- line 438: factors ... factors
- paragraph starting on line 443: the sentence is not clear.
- line 448: "one must one" maybe should read "one must not"?
- line 530: missing the word "previous"
- line 550-1: fix sentence: it does not read well.
- line 611: A label "E2" is mentioned to appear in fig. 8, but there is no such label in the figure.

Reviewer 2 ·

Basic reporting

In 6 experiments, Woods and colleagues investigated whether people prefer food items placed in odd rather than even number of elements. A very large sample size was collected, especially for the first experiments. Overall, they found no consistent evidence supporting the belief that it is better to present odd rather even numbers of items.
I believe that the article is written in clear academic English and includes sufficient introduction and background. Figures and graphs are also clear enough and relevant to the content of the article.

Experimental design

As far as I am aware, the submission describe original primary research. I think the authors clearly identified the theoretical gap investigated and how they tried to fill this gap. Methods are quite clear, and I cannot see any problem with ethical standards.

Validity of the findings

Results are not too much strong, but I know PeerJ also welcomes results that are not fully conclusive. Instead I am afraid the authors should include a control experiment before final decision.

In these experiments “3” was the odd number and “4” the even number. The authors correctly took into account the contribution of cumulative surface area of the stimuli. However, they did not properly take into account the role of numerosity. There is evidence that, in food choice tasks, non-human animals sometimes prefer to reach the group containing the larger number of food items (regardless to overall volume of the groups). See for instance, Hauser et al 2000 Proc R Soc Lon B and Uller & Lewis 2009 Anim Cogn.
In humans the relative contribution of number vs. continuous quantities is poorly known in food choice tasks. However, in everyday life the association between number and continuous quantities is unlikely to be violated, so it is possible that our species also prefer the larger number of food items in a relative numerosity judgment. If humans tend to prefer the larger number of items, it is likely that they would prefer the plate containing 4 items. So, null result here reported with respect to the ‘odd vs. even’ issue might be due to the fact that the authors contrasted to opposite preferences: 1) preference of odd items (3) and preference for the group containing the larger number of items (4).

This issue can be easily tackled by adding 2 control tests:
4 vs. 5 (this time 5 is also both the larger group and the odd one). We should expect a high preference for the group containing 5 items.
4 vs. 6 (to verify if humans do prefer the larger number of items).

A minor note: could you add the figure of stimuli in Exp. 3?

Additional comments

I think this might be a potentially useful contribution to the literature provided that the authors include additional data (see “Validity of the Findings” section).

---

## Round 0.2 · accepted · Accept

I've received feedback from on of the referees and I've read your reply letter and the revised manuscript. The requests of both referees have been handled carefully and satisfactorily, I think your manuscript has improved substantially and can be published as is.

·

Basic reporting

No Comments

Experimental design

No Comments

Validity of the findings

No Comments

Additional comments

I am happy with the review. I only found a few shortcomings:
- Lines 116-7 need editing: missing coma after ‘however’, and chose between “however…. though”.
- Lines 645-647: “where the odd numbered plate has fewer elements than the even numbered plate, and where the even numbered plate has more than the odd number”: isn’t that the same thing? By checking the pairs (lines 670-671), I see there are actually pairs in which odd is greater than even. So I think the sentence just needs to be fixed.
- Line 682: “and ASKED for”